# On Sound Scattering and Acoustic Properties of the Upper Layer of the Sea with Bubble Clouds

**Vladimir A. Bulanov \*, Lubov K. Bugaeva and Andrey V. Storozhenko**

V.I. Il'ichev Pacific Oceanological Institute, Far East Branch of Russian Academy of Sciences, 690041 Vladivostok, Russia; bugaeva@poi.dvo.ru (L.K.B.); storozhenko_and@poi.dvo.ru (A.V.S.)
\* Correspondence: bulanov@poi.dvo.ru

**Abstract:** The presence of bubbles near the sea surface under certain conditions leads to abnormal sound scattering and a significant change in the acoustic properties of the upper layer of the sea. The article presents some results of sound scattering studies under various sea conditions, up to stormy conditions, when extensive bubble clouds arise. By the method of unsteady acoustic spectroscopy, data on the size distribution of bubbles at various depths have been obtained, which can be described by a power function with exponential decay at small bubble sizes of the order of 10 microns. Estimates of the gas content in bubble clouds and their influence on the acoustic characteristics of the upper layer of the sea have been carried out. It is shown that at sufficiently high concentrations, sharp increases in absorption and dispersion of the sound velocity are observed. Modeling of sound propagation in the presence of a quasi-homogeneous bubble layer shows that it leads both to a change in the laws of the average decay of the sound field along the sound propagation path and to a change in the shallow spatial structure of the field.

**Keywords:** seawater; bubbles; plankton; sound scattering; sound attenuation; acoustic spectroscopy





## 1. Introduction

According to its hydro-physical characteristics, the upper layer of the sea sharply differs from the rest of the marine environment. The near-surface layer of the ocean water column changes significantly with strong winds and developed surface waves [1–9]. Under these conditions, it is characterized by developed turbulence, abnormally high concentrations of gas bubbles, gas saturation of water and a large gas exchange between the ocean and the atmosphere. [3–7,9–13]. Under these conditions, acoustic characteristics also change, such as the coefficients of scattering, absorption and dispersion of the speed of sound, the parameter of the nonlinearity of seawater, which become dependent on the presence of air bubbles formed when wind waves collapse [1–3,6–8,14–16]. Along with the change in these average characteristics of the near-surface layer, their fluctuations are sharply pronounced, which are associated with bubble structures that are born when wind waves collapse [4–8]. These structures, as a rule, affect the scattering and propagation of sound [14–16].

The aim of the work is to study the structure and dynamic characteristics of the upper layer of the sea saturated with gas bubbles, as well as the relationship between the acoustic characteristics of the agitated upper layer of the sea and the characteristics of bubble clouds formed by the collapse of surface waves in strong winds.

The measurement of the concentration of bubbles and their size distribution $g(R)$ in the sea was carried out by various methods (mainly optical and acoustic) and the results are presented in a large number of papers [1–14]. Nevertheless, many issues of the distribution of bubbles in depth and the regularities of the evolution of the $g(R)$ function over time after the passage of various disturbances, including the effects of the collapse of wind waves and their relaxation to an undisturbed state, remained unclear [10–13]. It is usually

assumed that bubble clouds arise after the collapse of surface waves in local regions of space and such clouds are distributed quite freely, so that the characteristic horizontal size of the cloud is less than the distance between neighboring clouds. However, at sufficiently high wind speeds, during the transition to stormy conditions, the distance between local clouds is reduced, the saturation of the upper layer with gas increases sharply and at the same time the lifetime of local bubble clouds increases, which often leads to the overlap of individual clouds and the formation of a continuous bubble layer inhomogeneous in space. The acoustic properties of such a layer remain poorly studied in many respects. One of the main problems in measuring the characteristics of the bubble layer under various meteorological conditions is the need to carry out long-term measurements with high spatial and temporal resolution [5,6]. Such measurements in shallow seas can be carried out using radiators installed on the bottom and directed vertically upwards. To implement such measurements, a set of emitters with different radiation frequencies is usually used in order to maximize the overlap of the spectral region of bubble structures. As a rule, narrow-beam linear acoustic emitters are used to locate clouds. There is also experience in the use of parametric acoustic emitters that allow scanning in the range of difference frequencies with a little varying pumping frequency and maintaining the directional characteristics in a wide range of difference frequencies [17,18]. An important problem is the separation of scattering on resonant bubbles from scattering on non-resonant bubbles and other heterogeneous and homogeneous inhomogeneities of the water column (plankton, suspensions, turbulence). The use of unsteady scattering [17,18] of sharply directed ultrasound beams using parametric acoustic emitters makes it possible to separate resonant scattering from the non-resonant background and obtain experimental data on the structure of bubble clouds formed when wind waves collapse and on their involvement in the sea thickness. The main objective of this work was to study the distribution of scattering coefficients, absorption and dispersion of the sound velocity of seawater in the near-surface layer of the sea in relation to the distribution of bubbles in the upper layer of the sea.

Using the nonstationary scattering method, we were able to identify the contribution to sound scattering caused individually by bubbles and other heterogeneous and homogeneous inhomogeneities in the aggregate (plankton, suspensions, turbulent layers, micro-flows and other homogeneous inhomogeneities). The theoretical foundations of the method are presented in Section 2.1.

With a slight sea disturbance and a low wind speed, bubbles formed after the collapse of wind waves are often localized near the sea surface, where their concentration prevails over the concentration of other inclusions. However, with increased excitement, bubbles often reach areas where the contribution of phyto- and zooplankton becomes significant. Section 3.1 below shows typical areas in which joint sound scattering on bubbles and plankton must be taken into account.. Under these conditions, separation of contributions to sound scattering from bubbles and plankton is required. Otherwise, inflated concentrations of bubbles can be obtained, as well as an incorrect distribution of the concentration of bubbles in depth. In this issue, the results presented by us are new, differing from the works of other authors [2–16], for whom such an analysis has not been carried out.

Additionally, it should be noted that it is also necessary to correctly assess the structure of the bubble layer in the near-surface layers, where, despite the absence of heterogeneous inhomogeneities, there may be a significant contribution of homogeneous inhomogeneities (turbulence, microstructure of temperature, salinity, etc.). Their combined effect leads to fluctuations in the refractive index, which directly leads to an additional background of sound scattering [1].

For the first time, a task of this kind was set by us and partially completed in [17], but this was described most fully in [18]. This paper presents the generalized results of time-consuming experiments on sound scattering under various hydrological and meteorological conditions with inverted radiators installed on the bottom of the sea, contained in Section 3.2. They allowed us to obtain experimental data on the bubble size distribution function, on the basis of which a new semi-empirical Equation was proposed. It should

be noted that the proposed Equation is one of a number that have also been used by other authors [2–16]. It seems to us that the proposed Equation clarifies the actual size distribution of bubbles in the sea and allows us, quickly and fairly, to fully assess the contribution of bubbles to the absorption coefficient and dispersion of the speed of sound in seawater. The theoretical foundations of the assessment in the framework of a homogeneous model are presented in Section 2.2. The experimental data obtained made it possible to estimate the variability of the total gas content and the specified acoustic characteristics in bubble clouds, which are contained in Section 3.3. Based on the data obtained, numerical modeling was carried out to study the influence of the near-surface layer of bubbles on the propagation of sound for shallow seas using the normal mode approximation contained in Section 3.4. In Section 4, a general discussion of the results is conducted.

## 2. Materials and Methods

### 2.1. Theoretical Foundations

The basis of the method for measuring the size distribution of $g(R)$ in the sea is based on sound scattering measurements [1–3,5,6]. The study of the backscattering of acoustic pulses in clouds of bubbles in a shallow sea often occurs with the use of inverted radiators installed on the bottom of the sea. Having information about the amplitude $P_i$ of the wave incident on the scattering volume $V$ and the amplitude $P_{bs}$ of the wave scattered by this volume in the opposite direction, it is possible, according to the monograph [1], Ch.9, to determine the volume scattering coefficient in the Born approximation in the form:

$$m_V = \frac{2}{\pi\theta^2 c\tau} \left( \frac{P_{bs}}{P_i} \right)^2 \tag{1}$$

where $\theta$ is the width of the emitter directivity characteristic, $c$ is the speed of sound, $\tau$ is the duration of the sound pulse, $P_i(r) \approx A\exp(-\alpha r)/r$, $A$ is the calibration value of the amplitude of the emitted sound, measured in Pascals and usually reduced to a distance of 1 m, when the sound absorption at a distance $r$ is small, and $\alpha$ is the sound absorption coefficient.

The value $P_{bs}$ is measured directly in the experiment. Using the repeated use of pulses, it is possible using Equation (1) to register the change in time of the average scattering coefficient of sound and its fluctuations in the entire thickness of the liquid with a high spatial resolution determined by the width of the directional characteristic of the emitter $\theta$ and the length of the acoustic pulse $l_{imp} = c\tau/2$, and a high temporal resolution determined by the time interval between the pulses of the emitted sound. If the calibration for the coefficient $A$ in Equation (1) is missing or has changed, its value can be restored based on the available information on sound reflection from the media interface. The main step is to determine the value of $A$ in ideal conditions of a calm sea without waves and near-surface bubbles. Under these conditions, we can state that $\alpha h << 1$ and $V \approx -1$. If we measure the pressure in the first reflected pulse, we can find the value $A$ in the form $A = P_h 2h$. Then $P_i(r) = P_h \exp(-\alpha r)(2h/r) \approx P_h(2h/r)$ and we finally get

$$m_V(r) = \frac{1}{2\pi c\tau\theta^2} \left( \frac{P_{bs}(r)}{P_h} \right)^2 \left( \frac{r}{h} \right)^2 \exp\left( \int_0^r \alpha(x)dx \right) \tag{2}$$

Differentiating (2) by $r$, one can obtain an Equation relating the sound absorption and the scattering coefficient

$$\alpha(r) = d/dr\left\{ \ln\left[ m_V(r)/\left( r^2 P_{bs}(r)^2 \right) \right] \right\} \tag{3}$$

The simplest bubble size distribution function $g(R)$ can be found by the frequency dependence of the sound scattering coefficient $m_V(\omega)$, assuming that the main contribution

to sound scattering is made by resonant bubbles whose radius $R(\omega)$ is related to the frequency $\omega$ according to the Minnert formula $R(\omega) = \sqrt{3\gamma P_0/\rho}/\omega$ [2,3,6]:

$$g(R(\omega)) = \frac{2\delta_\omega}{\pi R^3(\omega)} m_V(\omega) \tag{4}$$

where $\delta_\omega$ is the coefficient of resonant attenuation on frequency $\omega$, $P_0$ is hydrostatic pressure, $\gamma \approx 1.4$ is adiabatic constant of the gas inside the bubble, and $\rho$ is a density of the liquid.

In the presence of other non-resonant inclusions along with resonant bubbles, the total sound scattering coefficient from the unit volume of the medium $m_V$ can be written as [17,18]:

$$m_V = m_V{}^{(b)} + m_V{}^{(s)} = \int\limits_{\{R\}} \left[ \left| f^{(b)} \right|^2 g^{(b)}(R) + \left| f^{(s)} \right|^2 g^{(s)}(R) \right] dR \tag{5}$$

Here, $f^{(b)}$ and $f^{(s)}$ are the amplitudes of monopole (volumetric) stationary sound scattering on bubbles and non-resonant inclusions, andindices (*b*) and (*s*) indicate that the quantities belong to bubbles and non-resonant inclusions, respectively. Note that the function $g(R)$ is related to the number of bubbles per unit volume by the Equation $N = \int_{R_{\min}}^{R_{\max}} g(R) dR$. The expression for $f^{(b)}$ has the following form [1–3,6]

$$f^{(b)} = R / \left\{ \left[ \left( R_\omega{}^2/R^2 \right) - 1 \right]^2 + \delta_\omega{}^2 \right\} \tag{6}$$

Taking into account transients when the bubble is swinging at resonance leads to a dependence of $m_V{}^{(b)}$ on the pulse duration $\tau$ [18]:

$$m_V{}^{(b)}(\tau) = m_V{}^{(b)}(\infty) F(\tau/\tau_0), \; m_V{}^{(b)}(\infty) = (\pi/2) R^3 g^{(b)}(R)/\delta(R) \tag{7}$$

$$F(\tau/\tau_0) = 1 - [1 - \exp(\tau/\tau_0)]/(\tau/\tau_0), \; \tau_0 = 1/\omega\delta = Q/\omega. \tag{8}$$

Here $m_V{}^{(b)}(\infty)$ is the coefficient of stationary resonant scattering on bubbles. The function $F(\tau/\tau_0)$ determines the evolution of the cross-section of the non-stationary resonant scattering, and therefore it helps in practice to separate the resonant scattering from the non-resonant background, as well as to determine the Q-factor of bubbles at different frequencies according to Equation (8). The use of frequency-tunable directional emitters makes it possible to implement non-stationary acoustic spectroscopy of bubbles [18] in the form:

$$W^2(\tau) = \left( \pi c \theta^2/2 \right) \left[ m_V{}^{(b)}(\tau) + m_V{}^{(s)} \right], \tag{9}$$

$$g^{(b)}(R) = 4\delta(R) \left[ W^2(\infty) - W^2(0) \right] / \left( \pi^2 c \theta^2 R^3 \right). \tag{10}$$

where $W(\tau) = (1/\sqrt{\tau})(P_s/P_i)$. The scattering coefficient on the remaining inclusions can be determined by the Equation:

$$m_V{}^{(s)} = 2W^2(0) / \left( \pi^2 c \theta^2 R^3 \right). \tag{11}$$

The designations $W(\infty)$ and $W(0)$ correspond to the conditions $\tau >> \tau_0$ and $\tau << \tau_0$, respectively. Thus, the bubble size distribution function can be determined from the data of the inverse linear scattering of acoustic pulses of long and short duration.

### 2.2. Evaluation of Acoustic Characteristics Based on a Homogeneous Model of Water with Bubbles

To describe the acoustic properties of a micro-homogeneous fluid, effective acoustic parameters are often used, which are usually determined within the framework of a

homogeneous continuum model [1,7,15,19]. The effective density of a micro-homogeneous liquid can be determined using the Equations

$$\rho_e = \rho(1-x) + \rho'x, \; x = \frac{4}{3}\pi \int_{R_{\min}}^{R_{\max}} R^3 g(R)dR, \tag{12}$$

where $\rho$ and $\rho'$ is the density of the liquid and the contents of the bubble, respectively, and the strokes hereafter refer to bubbles. The effective compressibility of a micro-homogeneous liquid, defined as $\beta_e = (1/\rho_e)(d\rho_e/dP) \equiv (\rho_e)_P/\rho_e$, is equal to [19,20]

$$\beta_e = \beta + \frac{4}{3}\pi \int_{R_{\min}}^{R_{\max}} (\mathcal{K}-\beta)R^3 g(R)dR \equiv \beta + x(\mathcal{K}-\beta). \tag{13}$$

Here $\mathcal{K}$ denotes the compressibility of a single bubble in a liquid, which takes into account the resonant and relaxation responses of bubbles to the influence of an external force, and in general the value $\mathcal{K}$ is a complicated function of the size of the bubble and the frequency of sound $\mathcal{K} = \mathcal{K}(R, \omega)$ [18–20]. In the absence of these effects, the compressibility $\mathcal{K}$ is equal to the compressibility of the material of a single bubble, i.e., $\mathcal{K} = \beta'$. Taking into account the polydispersity of the bubble mixture in general, the designations $x(\mathcal{K}-\beta)$ hereafter should be understood in an integral sense, as a result of the effect of the integral operator $x$ on the function $(\mathcal{K}-\beta)$, leading to an expression $(4\pi/3)\int_{R_{\min}}^{R_{\max}} (\mathcal{K}-\beta)R^3 g(R)dR$ that takes into account the size distribution of bubbles. With a monodisperse distribution $g(R) \sim \delta(R - \overline{R})$, the value $x$ is the usual volume concentration $x = (4\pi/3)\overline{R}^3 N$, where $N$ is the number of bubbles of the same radius $\overline{R}$ per unit volume of the liquid. The following relation between $g(R)$ and the concentration of bubbles per unit volume is $N = \int_{R_{\min}}^{R_{\max}} g(R)dR$, where $g(R) = dN/dR$. The compressibility of a single bubble in a liquid $\mathcal{K}$ takes into account the resonant and relaxation responses of bubbles to the influence of an external force and, in general, the value $\mathcal{K}$ is a complex function of the size of the bubble and the frequency of sound $\mathcal{K} = \mathcal{K}(R, \omega)$ [18–20]. In the absence of these effects, the compressibility $\mathcal{K}$ is equal to the compressibility of the material of a single bubble, i.e., $\mathcal{K} = \beta'$. The effective velocity of sound in a liquid with bubbles $\widetilde{c}_e$ can be calculated based on a generalization of Wood's formula [19,20]

$$\widetilde{c}_e = \frac{1}{\sqrt{(\rho_e)_P}} = \frac{1}{\sqrt{\rho_e \beta_e}}, \tag{14}$$

which formally looks like the well-known Wood formula, but with modified effective parameters $\rho_e$ and $\beta_e$ in accordance with (12) and (13). The real part $c_e = \mathrm{Re}(\widetilde{c}_e)$ determines the phase velocity of the pressure wave in the form:

$$c_e = c\,\mathrm{Re}\left\{ \left[1 + \frac{4}{3}\pi \int_{R_{\min}}^{R_{\max}} \left(\frac{\mathcal{K}-\beta}{\beta}\right)R^3 g(R)dR\right][1-x]\right\}^{-1/2}. \tag{15}$$

The attenuation coefficient $\alpha$ of a wave propagating in a liquid with bubbles can be determined using the wave number $\alpha = \mathrm{Im}k_e = \omega\mathrm{Im}(1/\widetilde{c}_e)$ and Equation (3) in the form:

$$\alpha = \mathrm{Im}(k_e) = (\omega/c)\mathrm{Im}\left[1 + \frac{4}{3}\pi \int_{R_{\min}}^{R_{\max}} \left(\frac{\mathcal{K}-\beta}{\beta}\right)R^3 g(R)dR\right]^{1/2}. \tag{16}$$

Taking into account the resonant properties of bubble compressibility $\mathcal{K} = \mathcal{K}(R, \omega)$, the effective sound velocity $c_e$ and sound absorption coefficient $\alpha$ can be calculated using Equations (15) and (16) with parameters obtained using experimental data at low concentrations $x$:

$$c_e \approx c\,\mathrm{Re}\left(1 - \frac{2\pi}{3}\frac{\rho c^2}{\gamma P_0}\int_0^{\infty} \frac{g(R)dR}{q(R, R_\omega)}\right), \tag{17}$$

$$\alpha \approx \frac{\omega}{c}\text{Im}\left(\frac{2\pi}{3}\frac{\rho c^2}{\gamma P_0}\int_0^\infty \frac{g(R)dR}{q(R, R_\omega)}\right), \tag{18}$$

where $q(R, R_\omega) = 1 - (R/R_\omega)^2(1 + i/Q_\omega)$, $Q_\omega = 1/\delta_\omega$ is the Q-factor of a bubble of radius $R_\omega$.

### 2.3. Experimental Methods and Equipment

Experimental work was carried out in the shallow sea. In Vityaz Bay of the Sea of Japan (100 km south of Vladivostok), a bottom system with sonar emitting and receiving antennas (depth of 12 m) was installed for the purpose of long-term study of acoustic characteristics in the sea under various hydrometeorological conditions (Figure 1).

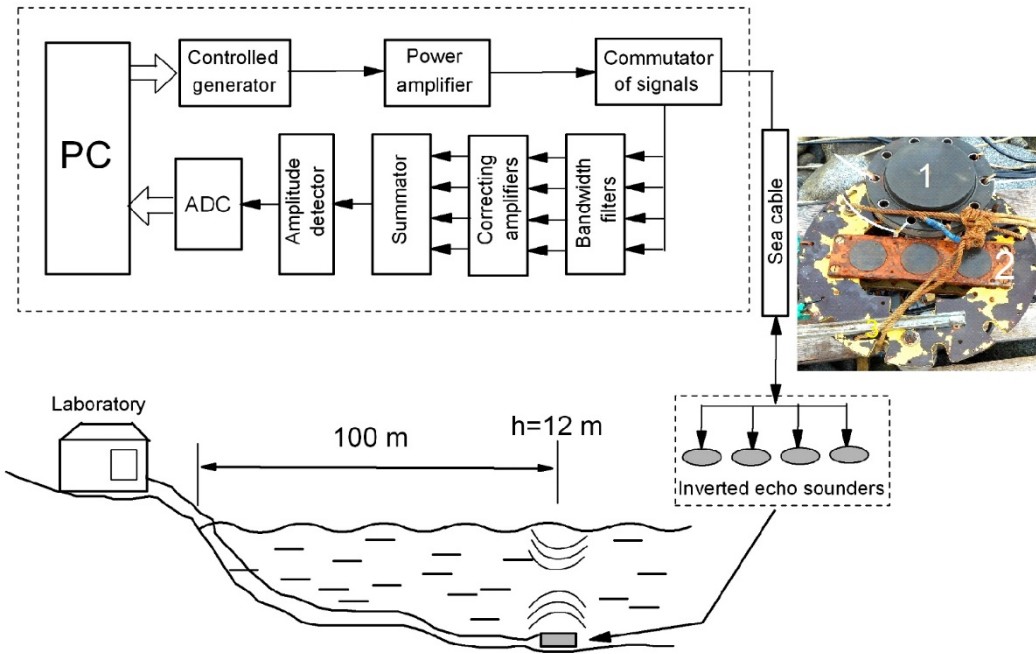

**Figure 1.** The scheme of acoustic measurements from the bottom station. The inset shows the appearance of the bottom station emitters after lifting out of the water: 1—a radiator with a frequency of 145 kHz, 2—a three–element emitter with frequencies of 138, 216, 519 kHz.

The acoustic system for measuring sound scattering included a sound emission path with different frequencies, piezoceramic converters, a reception path and a system for input and primary processing of acoustic information. The system of input and primary processing of acoustic information included a 14-bit interface input board from the company "Rudnev and Shilyaev" La2-USB with a maximum quantization frequency of 400 kHz, an interface 12-bit input board from the company "L-Card" E20-10 with a maximum quantization frequency of 10 MHz, personal computers and special programs for processing and visualization of acoustic signals.

One of the important components of the system was a three–element emitter having the width of the main lobe of the directivity characteristic at a frequency of 138 kHz equal to 11.5°, at a frequency of 216 kHz is 7.2°, at a frequency of 519 kHz is 3°. As a parametric emitter at pumping frequencies of 200 kHz at difference frequencies of 15–40 kHz, the FURUNO emitter type CA200-8B (Japan) was most often used, with an operating frequency of 200 kHz and a maximum permissible power of 2 kW. The width of the radiation pattern at the operating frequency is 5.6°. An emitter with a frequency of 145 kHz was also often used, with directivity of 5°, and power 2 kW.

The coastal equipment allowed for multi-frequency measurement of scattering signals by various methods. The method of simultaneous emission of pulses of different frequencies was used, followed by filtering of the received signals through channels [21] (Figure 1).

The programmable generator GSPF-053 from the company "Rudnev and Shilyaev" (Moscow) was used as a digital signal generator. Broadband power amplifiers of the U7-5 type were used as pre-amplifiers. Terminal amplifiers were made on the basis of high-voltage transistors and allowed to raise the output voltage to 400 volts. The signal switch was made according to the scheme of diode switches of echo sounders. For narrowband filtering and amplification, selective nanovoltmeters SN-233 and SN-232 from UNIPAN (Poland), third-octave filters RFT01018 from Robotron (Germany), microphone amplifiers RFT00011 from Robotron (Germany), filters for individual frequencies were manufactured in the laboratory of hydrophysics of the POI FEB RAS. The coastal equipment was located in the immediate vicinity of the bottom station and was connected by an underwater cable (Figure 1).

Table 1 presents the main parameters of the typical experimental equipment for conducting experiments.

**Table 1.** Main parameters of experimental equipment.

| Type | Main Characteristics |
| --- | --- |
| Acoustic emitters | 1. Three-element multi-frequency emitter: Frequency 138 kHz, directivity 11.5°, frequency 216 kHz, directivity 7.2°, frequency 519 kHz directivity 3°, power 2 kW<br>2. Emitter CA200-8B: 200 kHz, directivity 5.6°, power 2 kW.<br>3. 145 kHz emitter: directivity 5°, power 2 kW. |
| Equipment for the emission of acoustic signals | 1. Digital signal generator GSPF-053: 12 digits, frequency up to 10 MHz, voltage 5 V.<br>2. Broadband power amplifier U7-5: frequency up to 3 MHz, power 20 W, voltage 20 V<br>3. BIP power amplifier (POI): frequency up to 1 MHz, power 600 W, voltage up to 400 V |
| Equipment for receiving acoustic signals | 1. Selective nanovoltmeter SN-233: standard frequency up to 150 kHz, extended frequency up to 200 kHz; selectivity: (a) wide band, (b) 20 dB/octave, (c) 36 dB/octave, (b) 54 dB/octave; maximum gain $10^6$.<br>2. Third-octave filters RFT01018 (Robotron): frequency up to 200 kHz, switchable selectivity: (a) octave, (b) 1/3 octave; maximum gain 10.<br>3. Selective filters for individual frequencies (POI): standard frequencies 138 kHz, 145 kHz, 216 kHz, 512 kHz, with the possibility of frequency adjustment, Q factor 10.<br>4. Brüel & Kjærtype 2650 amplifier: frequency up to 200 kHz, LF and HF filters: 30 kHz and 2 kHz, Maximum gain $10^2$<br>5. Microphone amplifier RFT00011 (Robotron): frequency up to 1 MHz, maximum gain $10^6$ |
| Equipment for recording acoustic signals | 1. La2 USB ADC (Rudnev and Shilyaev): quantization frequency up to 400 kHz, 14 digits, 16 channels, maximum gain $10^3$.<br>https://rudshel.ru/show.php?dev=35, accessed on 23 May 2022.<br>2. ADC E20-10 (L-Card): quantization frequency up to 10 MHz, 12 digits, 4 channels, maximum gain $10^2$.<br>https://www.lcard.ru/products/external/e20-10, accessed on 23 May 2022. |

## 3. Results

### 3.1. Sound Scattering in the Near-Surface Layer of the Sea

A typical distribution of the volume scattering coefficient of sound $m_V(z, t)$ obtained using the bottom system is shown in Figure 2. The tidal fluctuations of the sea level are clearly visible. The night period is marked in gray color on the time axis.

Sound scattering is often displayed as a layer strength and recorded in logarithmic form according to the Equation $S_V = 10 \lg < m_V >$, where $< m_V > = (1/h) \int_h m_V(z) dz$, at the same time, the dimension of $m_V$ is taken in 1/m. Figure 3 shows the dependence of the average depth value on time in the entire water layer.

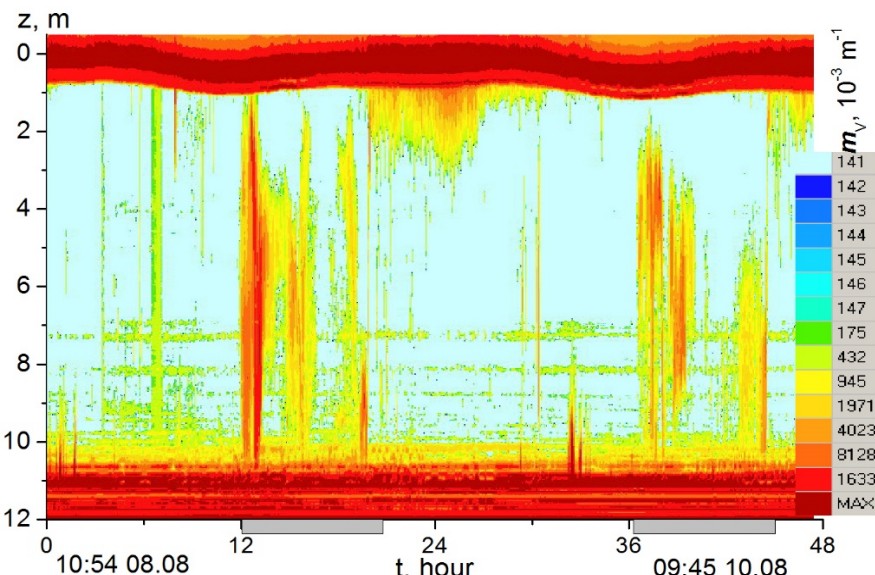

**Figure 2.** Changes in the volume scattering coefficient of sound $m_V(z,t)$ at a frequency of 138 kHz for two days from 10:54 on 8 August to 09:45 on 10 August 2017.

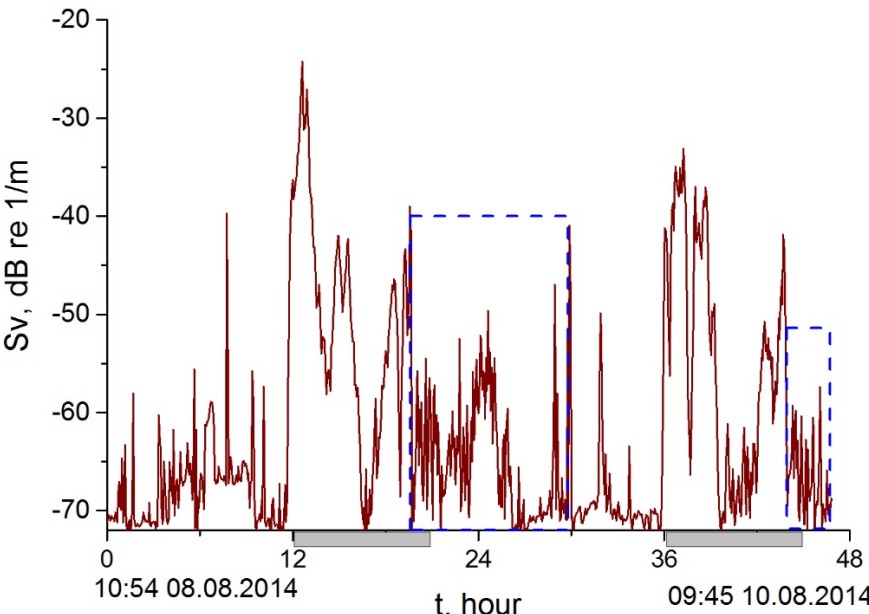

**Figure 3.** Change in the average sound scattering coefficient $S_V(t)$ for two days. The areas where the main contribution is made by bubbles are highlighted in gray color.

It can be seen from Figure 3 that during the night period the strength of the $S_V$ layer increases sharply, which is primarily due to diurnal migrations of plankton. The increased values in the daytime are associated with bubble clouds, the time intervals of which are highlighted with a dashed line in Figure 3. Sound scattering on the shelf is largely of biological origin and is almost an order of magnitude higher at night compared to daytime values. Figure 4 presents a detailed picture of sound scattering at a frequency of 145 kHz on near-surface bubble clouds and simultaneous sound scattering on plankton communities involved in the dynamics of internal waves. At the top and right of the figure are horizontal and vertical profiles of the sound scattering coefficient $m_V$ along the lines shown in Figure 4. A more detailed picture of the wave dynamics of plankton is presented in the upper inset. It can be seen that a powerful layer of bubbles is observed near the surface, extending to a depth of about 4–5 m.

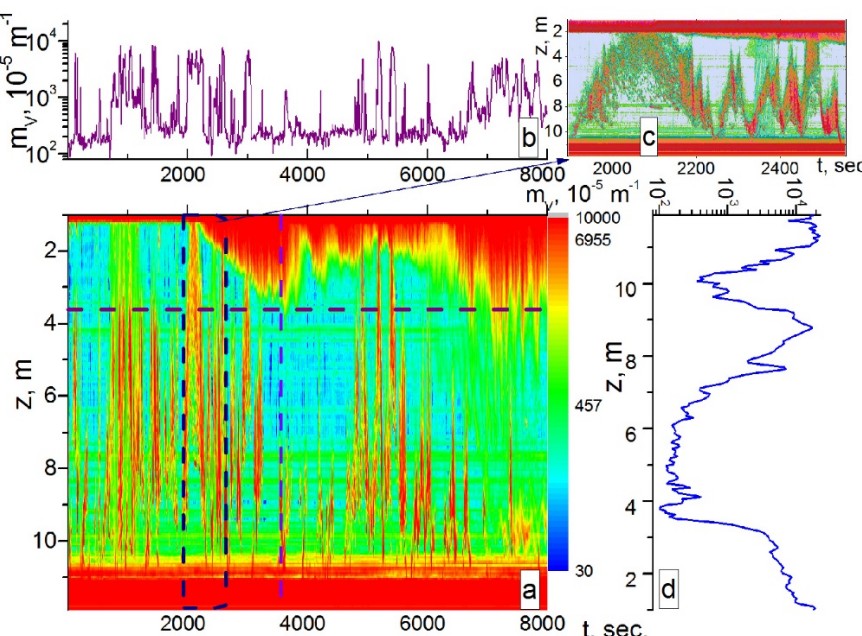

**Figure 4.** Sound scattering on near–surface clouds of bubbles and simultaneous sound scattering on zooplankton: (**a**) general view $m_V(z,t)$, (**b**) time profile of the sound scattering coefficient, - function $m_V(z_h,t)$ for depth = 3.5 m (horizontal line on part **a**), (**c**) vertical profile of the sound scattering coefficient $m_V(z,t_p)$, function for time $t_p$ = 3500 s from the beginning recordings (vertical line at $t_p$ = 3500 s on part **a**); (**d**) shows a typical modulation caused by internal waves of the sound scattering coefficient on plankton in a rectangular area highlighted by dotted lines on part (**a**).

Figure 5 shows the most typical results during 2 days of continuous recording of scattering on the shroud of bubbles involved in the water column by wind stresses for various sound frequencies: (a) 138 kHz, (b) 216 kHz, (c) 519 kHz.

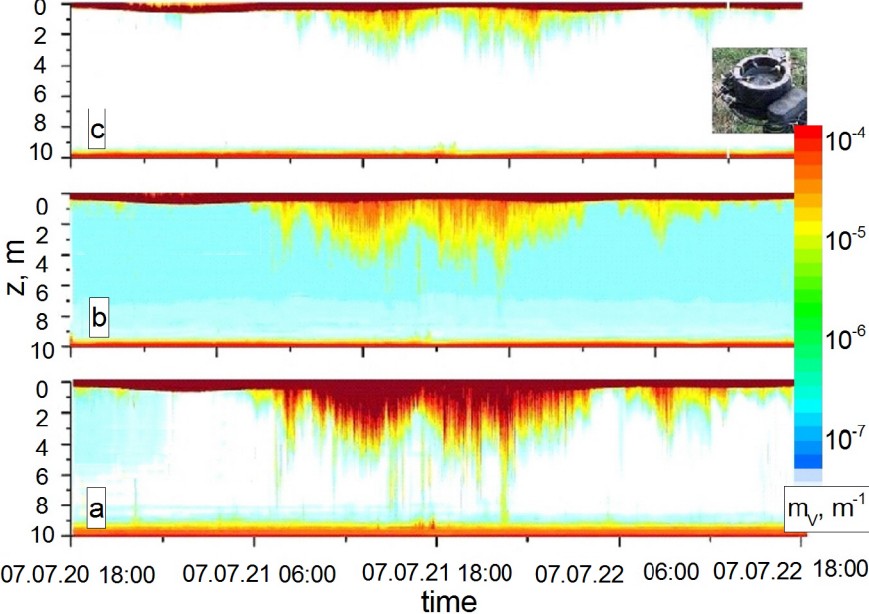

**Figure 5.** Variations of the sound scattering coefficient $m_V(z,t)$ caused by bubble clouds during 2 days. at frequencies (**a**) 138 kHz, (**b**) 216 kHz, (**c**) 519 kHz.

Figure 6 shows a detailed recording of the sound backscattering signals from the bottom station at a frequency of 138 kHz. Variations of sound scattering caused by air bubbles involved in the sea water column to a depth of 5–7 m are clearly visible.

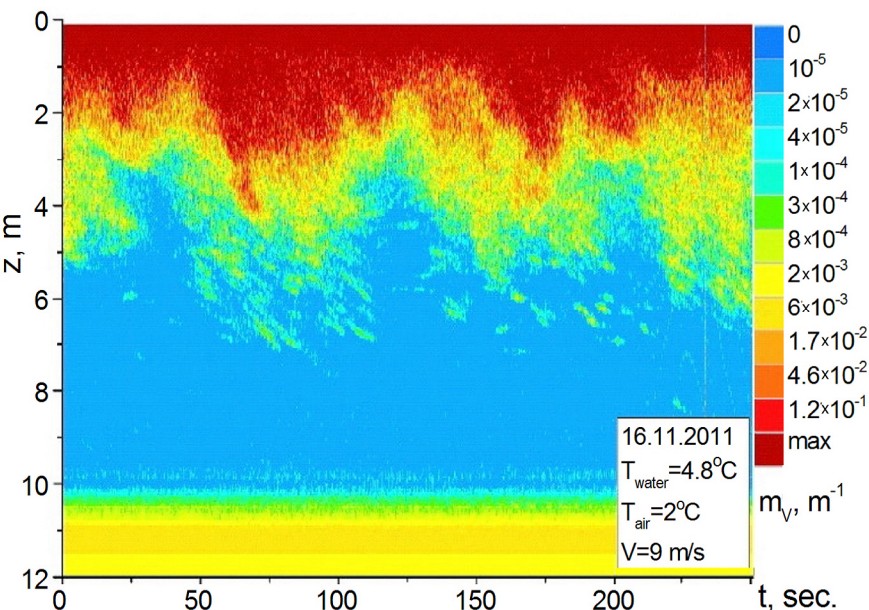

**Figure 6.** A typical result of an experiment on sound scattering from a bottom station at a frequency of 138 kHz at a high sounding frequency in order to study the near-surface layer of bubbles formed during the collapse of surface waves.

### 3.2. Bubble Size Distribution Function

The obtained data on sound scattering using Equations (4) and (10) allow us to obtain bubble size distribution functions. Instead of the value $g(R)$ having dimension ($\text{cm}^{-4}$), the value $N(R)$ is often used. It has a dimension ($\text{m}^{-3}\text{mkm}^{-1}$), which is associated with the $g(R)$ relation $N(R) = 10^2 g(R)$. At a fixed frequency of 138 kHz, which corresponds to a resonant size of 23 microns, the value of $N(R)$ resonant bubbles is shown in Figure 7 for different depths during the change of surface waves.

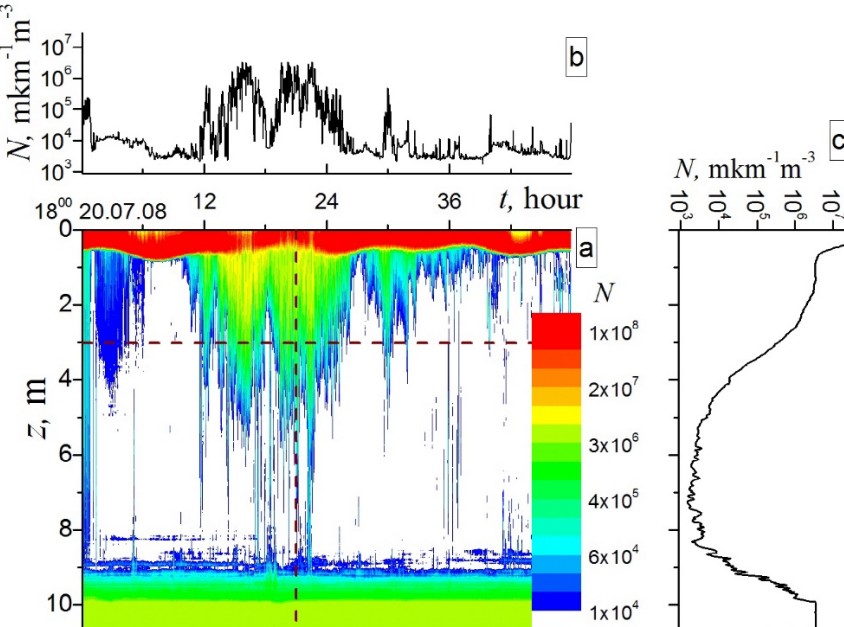

**Figure 7.** Distribution of bubbles resonant at a frequency of 138 kHz in depth and its change over two days due to fluctuations in wind speed (4–14 m/s) and sea waves: (**a**) general view; (**b**) change in time of bubble concentration at a depth of 3 m (indicated by a horizontal dashed line), (**c**) the change in the concentration of bubbles in depth at the time point 21 h after the start of the experiment (indicated by a dashed vertical line).

Experimental data on sound scattering in a wide frequency band using (10) allowed us to obtain bubble size distribution functions, which for different depths are shown in Figure 8 for different sea conditions: before a storm, during a storm and after a storm.

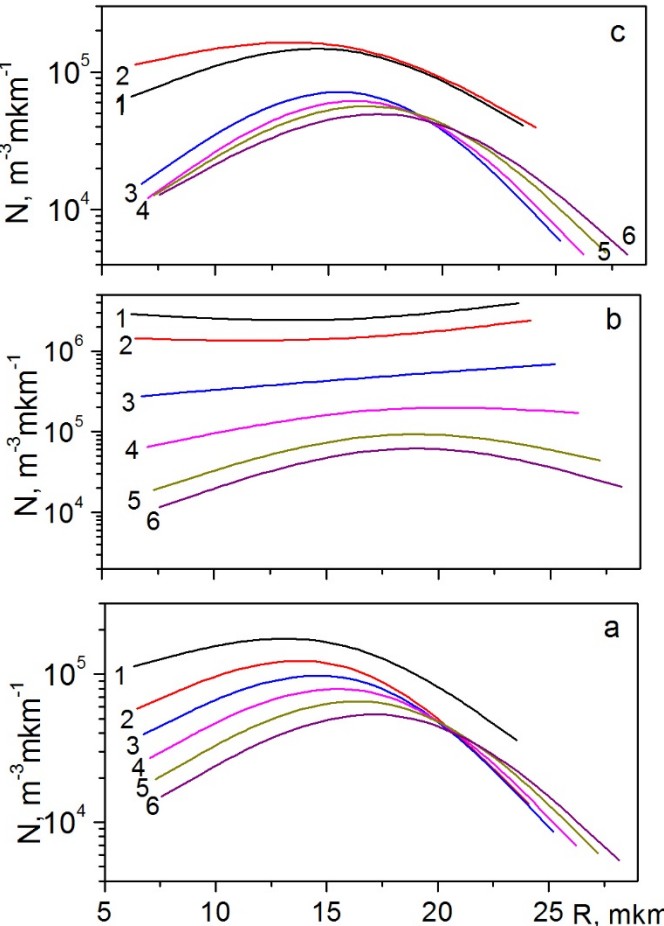

**Figure 8.** Bubble size distribution function at different depths (1–0.5 m depth, 2–1 m depth, 3–2 m depth, 4–3 m depth, 5–4 m depth, 6–5 m depth) during different periods of storm development: (**a**) before the storm 20 July 2008, (**b**) in a storm on 21 July 2008, and (**c**) after a storm on 22 July 2008.

It can be seen from Figure 8 that during periods without a storm, the maximum of the function $g(R, z)$ is observed, which is located at $R_p > 10$ microns, while the value $R_p$ depends on the depth. At $R > R_p$, there is a power dependence of the bubble size distribution function with an exponential decline with depth. Taking into account the decline of the function $g(R)$ at small $R$, the presence of a maximum at $R = R_p$ and limiting the spectrum from above to the maximum bubble size $R_m$ allows us to represent the bubble size distribution in an analytical form by the following Equation [14]:

$$g(R, z) = A_g R^{-n(z)} \exp\left[-m\left(\frac{R_p}{R} + \frac{R}{R_m}\right)\right] e^{-z/L} \tag{19}$$

where $L \sim a_L \cdot 10^{-3} U_{10}^{2.5}$. Here $L$ is given in meters, the coefficient $a_L$ is estimated by various authors [4,11,12,14] in the interval of $a_L = 2 \div 4$, while the wind speed $U_{10}$ at an altitude of 10 m is given in m/s), the indicator $m$ depends on the state of the sea and $m = 1 \div 3$. For moderate and calm waves $m \approx 1$. The advantage of such a record $g(R)$ is the practicality and speed of calculations of various parameters of the environment. It is also important that the exponent $n\sim3.3$ and the critical dimensions $R_p$, $R_m$ are natural parameters that follow from the Garrett–Lee–Farmer theory (GLF) [22]. Measurements of $g(R)$ on a large factual material under similar conditions of moderate sea conditions give

values in the interval $n \approx 3.3 \div 3.8$ [3–6,17,18], which is close enough to the estimate of $n \sim 3.3$ obtained for the inertial interval between the sizes $R_p$, $R_m$, following from the theory of GLF [22].

So, it is shown that a weakly perturbed structure is characterized by the presence of $g(R)$ with a maximum, the position of which varies depending on the depth. A completely different picture is observed during a storm—here a large number of both large and small bubbles are formed in the near-surface layers in the absence of a visible maximum, which, nevertheless, is available for bubbles located in the water column with depths greater than 3 m.

### 3.3. Estimates of Gas Content and Sound Absorption from Experimental Data

The assessment of acoustic characteristics in seawater bubble clouds can be carried out on the basis of experimental data obtained by sound scattering on bubble structures according to Equations (17) and (18) taking into account Equation (19).

Figure 9 shows the concentration dependence of the dimensionless sound velocity of water with gas bubbles $c_e(x)/c$ at $T = 20\ ^\circ$C, calculated for different sound frequencies in the case of a polydisperse mixture of bubbles for a power function of the bubble size distribution $g(R)$ according to (19) at $n = 3.8$, $m = 1$, $z = 0$ in the size range from $R_{\min} = 0.1$ mkm to $R_{\max} = 2000$ mkm. Figure 9 shows that in the concentration range $x$ from $10^{-6}$ to $10^{-5}$ there is a sharp decrease in the speed of sound in a liquid with bubbles.

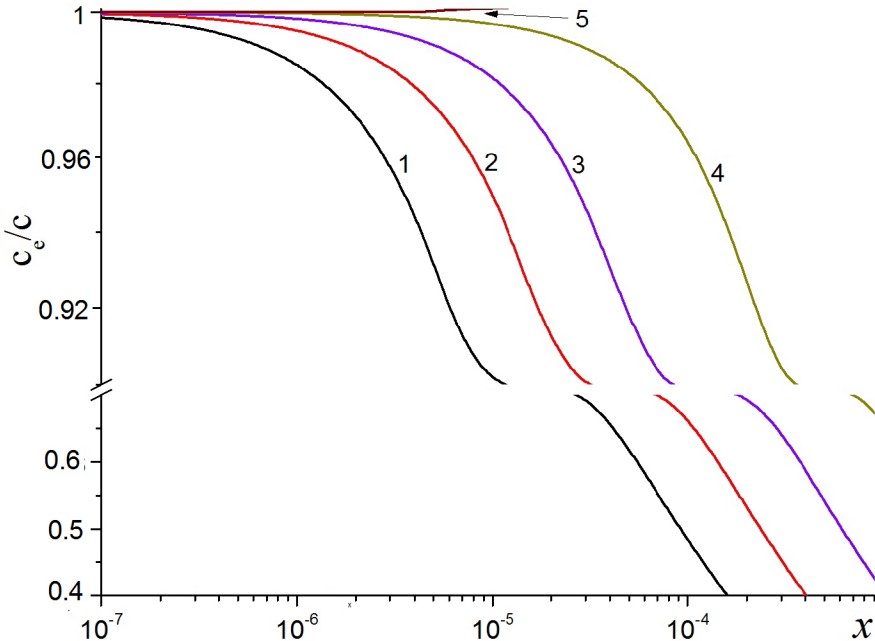

**Figure 9.** The dependence of the speed of sound in water with a polydisperse mixture of bubbles on the concentration at different frequencies: 1 kHz (1), 10 kHz (2), 100 kHz (3), 1 MHz (4), 7.5 MHz (5).

Figure 10 shows the frequency dependence of the sound absorption coefficient $\alpha$ in water with bubbles at $T = 20\ ^\circ$C, calculated by Equation (18) for a polydisperse mixture of bubbles at different volume concentrations of gas in bubbles $x = (4\pi/3) \int_0^\infty R^3 g(R) dR$, taking into account Equation (12) and $g(R)$ according to (19) at $n = 3.8$ as shown in the paper [20]. Here also is the frequency dependence of the sound absorption coefficient in fresh water $\alpha_0(f)$ and seawater $\alpha_{\text{sea}}(f)$ at $T = 20\ ^\circ$C and salinity 35 ppm. The frequency dependence of the sound absorption coefficient in fresh water $\alpha_0(f) \approx 2.3 \cdot 10^{-14} f^2$ (where $\alpha$ is in 1/m, $f$ is in Hz) and the frequency dependence of the sound absorption coefficient in seawater according to the Marsh-Shulkin formula are presented in monograph [1]. It can be seen that, in water with bubbles, the frequency dependence $\alpha(f)$ is weakly expressed. Such a weak frequency dependence is associated with the well-known predominant mechanism

of resonant attenuation in a cloud of bubbles with a wide bubble size distribution function $g(R)$ [1]. It should be noted that at high frequencies with small concentrations of bubbles less than $x_b \sim 10^{-8}$, the contribution to sound absorption in pure seawater without bubbles may prevail over the contribution from bubbles. In fresh water, the specified threshold $x_b$ is reduced by an order of magnitude. At concentrations of bubbles in the near-surface layers of seawater $x \sim 10^{-6} - 10^{-5}$ in conditions of developed excitement, the attenuation of sound will be entirely determined by the structure of the bubble cloud.

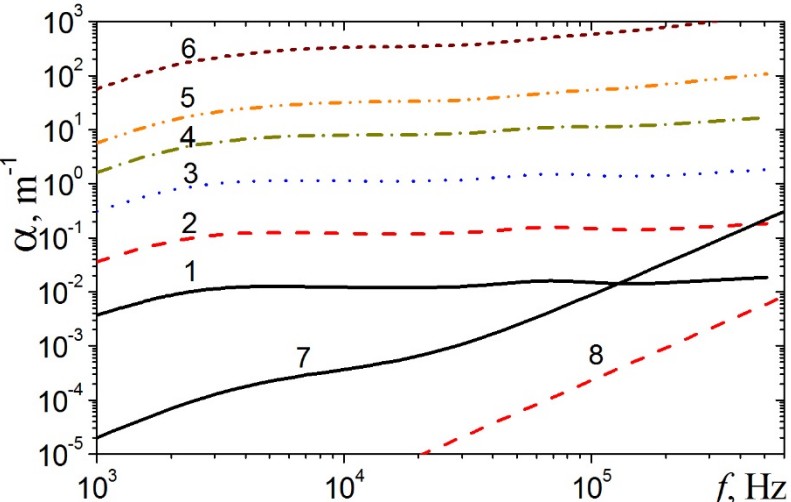

**Figure 10.** Frequency dependence of the sound absorption coefficient $\alpha$ in water at a temperature of 20 °C with a polydisperse mixture of bubbles at different gas concentrations in the bubbles $x$: 1—$x = 10^{-8}$, 2—$x = 10^{-7}$, 3—$x = 10^{-6}$, 4—$x = 10^{-5}$, 5—$x = 10^{-4}$, 6—$x = 10^{-2}$, 7—$x = 0$, sea water, 8—$x = 0$, fresh water.

In real conditions, as can be seen from the data in Figures 9 and 10, the concentration of bubbles varies greatly over time. Along with this value, the sound absorption coefficient $\alpha$ in the bubble layer changes greatly. Its value was calculated on the basis of experimental data obtained by sound scattering on bubble structures according to Equation (18). Figure 11 shows, as a typical example, calculations for sound absorption at a frequency of 145 kHz in the near-surface layer of bubbles at wind speeds from 9 to 13 m/s for about a day.

Figure 11 shows a significant excess absorption of sound in the bubble layer. Near the sea surface, sound absorption is more than 100 times higher than sound absorption in clear water $\alpha_0 \approx 10^{-6}$ 1/m. At great depths, sound absorption tends to the value of sound absorption in clear water. There is a significant variability in the thickness of the layer by the sound absorption coefficient, varying from 7 m to 1.5–2 m.

Important for practical applications is information about the gas content and structure of bubble clouds arising from the collapse of surface waves. Using the data for sound scattering, it is possible to calculate the bubble size distribution function, and then, as well as for sound absorption, shown above in Figure 11, to obtain data on the change in time of the average volume concentration of the gas $x = (4\pi/3) \int_{\{R\}} g(R) R^3 dR$ contained in the bubbles in the entire thickness of the seawater layer. We regularly carried out measurements of near-surface sound scattering associated with bubble structures. A typical change in time of the average volume concentration of gas contained in bubbles in the entire thickness of the sea water layer at a wind speed varying from 9 to 13 m/s is shown in Figure 12a. The spectrum of the gas concentration function is shown in Figure 12b. It can be seen that the concentration of bubbles is quite large, while it should be noted a significant variability in time of the average concentration of gas contained in the bubbles, which is due to the periodic collapse of wind surface waves and the formation of bubble clouds that reach a depth of 7–8 m. Characteristic spectral peaks are visible, corresponding to periods of

wind amplification above the sea surface and leading to abnormal sound absorption with a change in the $g(R)$ function, the details of which are visible in Figure 11.

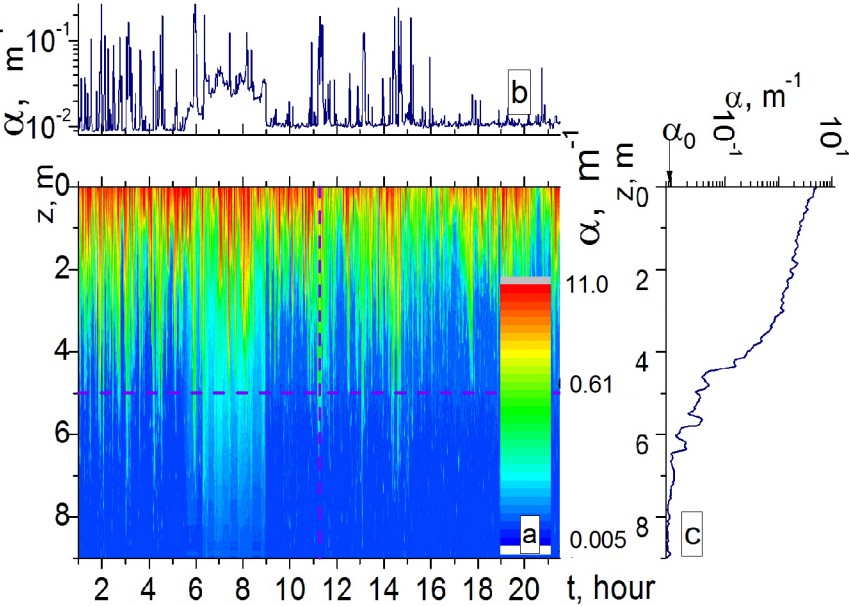

**Figure 11.** Changes in time of the sound absorption coefficient $\alpha(t,z)$ at a frequency of 145 kHz in the near-surface layer of bubbles at wind speeds from 9 to 13 m/s: (**a**) general view $\alpha(t,z)$, (**b**) change in time of the absorption coefficient at a depth of 5 m (indicated by a horizontal dashed line), (**c**) vertical section $\alpha(t_P,z)$ at time $t_P$ = 11.3 h from the beginning of the experiment (indicated by a dashed vertical line).

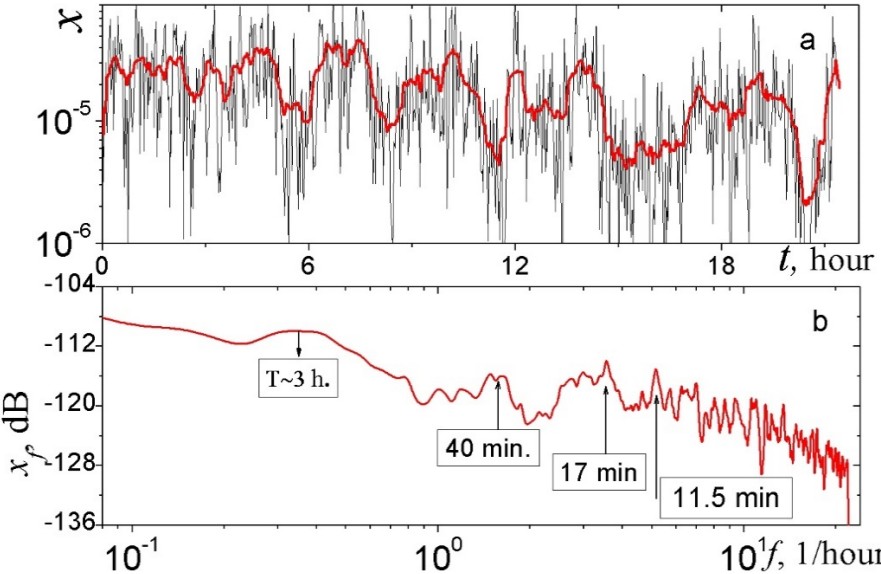

**Figure 12.** Variability in time of the average volume concentration of gas $x(t)$ contained in bubbles (**a**) and the spectrum of the gas concentration function (**b**) in the presence of bubble clouds formed by the collapse of wind waves.

### 3.4. On the Propagation of Sound in the Presence of Near-Surface Bubble Clouds

To study the influence of the near-surface layer of bubbles on the propagation of sound, numerical modeling was carried out for a shallow sea using the approximation of normal modes. For ease of analysis, a model of the simplest horizontally homogeneous, bubble–free isoscale underwater sound channel with absolutely reflective boundaries was chosen (the upper boundary is soft, the lower one is hard). The sound pressure is represented as the

sum of normal modes. The additional attenuation caused by the presence of a bubble layer is described by the imaginary part of the eigenvalues of the modes. Calculations of the sound field were performed using the KRAKENC program [23] for interacting modes. The thickness of the bubble layer was chosen equal to 7 m. The emitter with a frequency of $f$ = 1 kHz was located at a depth of 10 m at a sea depth of 42 m.

Figure 13 shows a 2D image of the acoustic field for different concentrations of bubbles in the near-surface layer. The calculations in Figure 13 show a strong change in the structure of the acoustic field when the concentration of bubbles exceeds $10^{-7}$. For this concentration of bubbles, the acoustic field in the bubble layer near the surface attenuates at a distance of about 400 m. The result is particularly impressive for a concentration of $10^{-6}$. Here, the field near the surface fades already at a distance of about 100 m. At the same time, the overall structure of the acoustic field in the thickness of the waveguide changes dramatically.

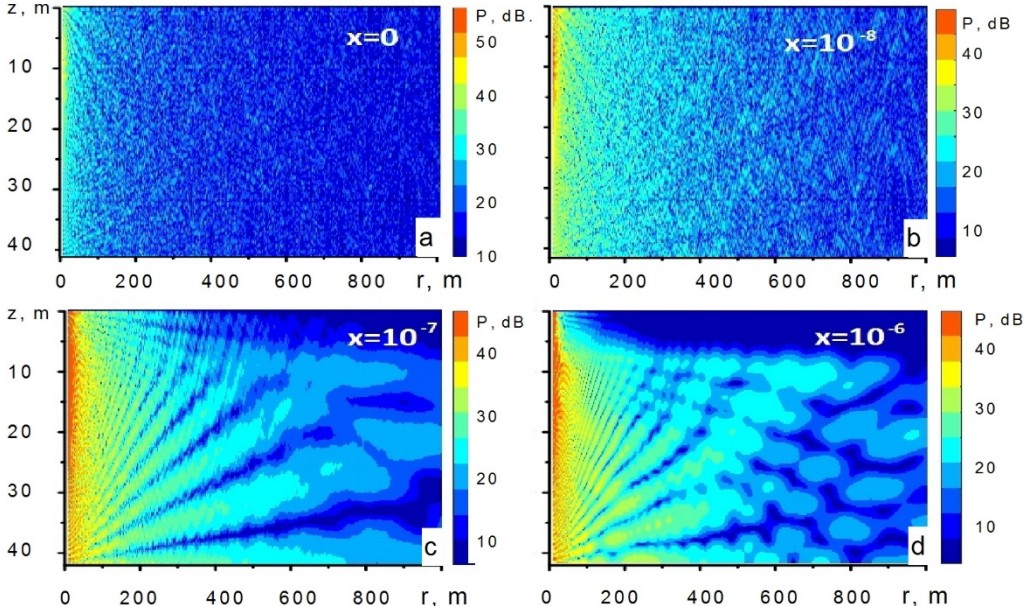

**Figure 13.** Acoustic field $|p(r,z)|$ of a source with a frequency $f$ = 1 kHz at a depth of 10 m in a channel with a near-surface layer of bubbles 7 m thick at different concentrations of bubbles: (**a**) $x = 0$, (**b**) $x = 10^{-8}$, (**c**) $x = 10^{-7}$, (**d**) $x = 10^{-6}$.

Figure 14 shows the dependences on the distance of the acoustic field pressure $|p(r,z)|$ at different concentrations of bubbles. Thick lines show the average depth pressure of the acoustic field. Figure 14 shows a strong dependence of the field decline on the distance at different depths. However, at large distances, the nature of the exponential decline of the field turns out to be close for various depths, including the average field. At these large distances, the main contribution to the energy of the field is made only by those components that do not interact strongly with the bubble layer, and therefore the sound absorption coefficient decreases sharply and approaches the value of the absorption coefficient in water without bubbles.

As a characteristic of the decline of the acoustic field with distance, we can take an expression for the average depth of the field of the form $P(r) = <|p(r,z)|>_z = (1/h)\int_0^h |p(r,z)|dz$, where $h$ is the depth of the channel. Then $P(r)$ it can be written in the form $P(r) = A\exp(-\alpha r)/\sqrt{r}$ according to which the attenuation coefficient of the sound can be calculated. The results of calculating the coefficients $\alpha$ show the following values $\alpha$: $\alpha = 1.5 \times 10^{-6}$ 1/m at $x = 0$; $\alpha = 9.5 \times 10^{-4}$ 1/m at $x = 10^{-8}$; $\alpha = 7.4 \times 10^{-3}$ 1/m at $x = 10^{-6}$. For comparison, the absorption coefficient of a plane sound wave $\alpha_b$ at a frequency of 1 kHz for the case of a homogeneous veil of bubbles in seawater has the following values: $\alpha_b = 1.5 \times 10^{-6}$ 1/m at $x = 0$; $\alpha_b = 3.5 \times 10^{-3}$ 1/m at $x = 10^{-8}$; $\alpha_b = 0.32$ 1/m at $x = 10^{-6}$.

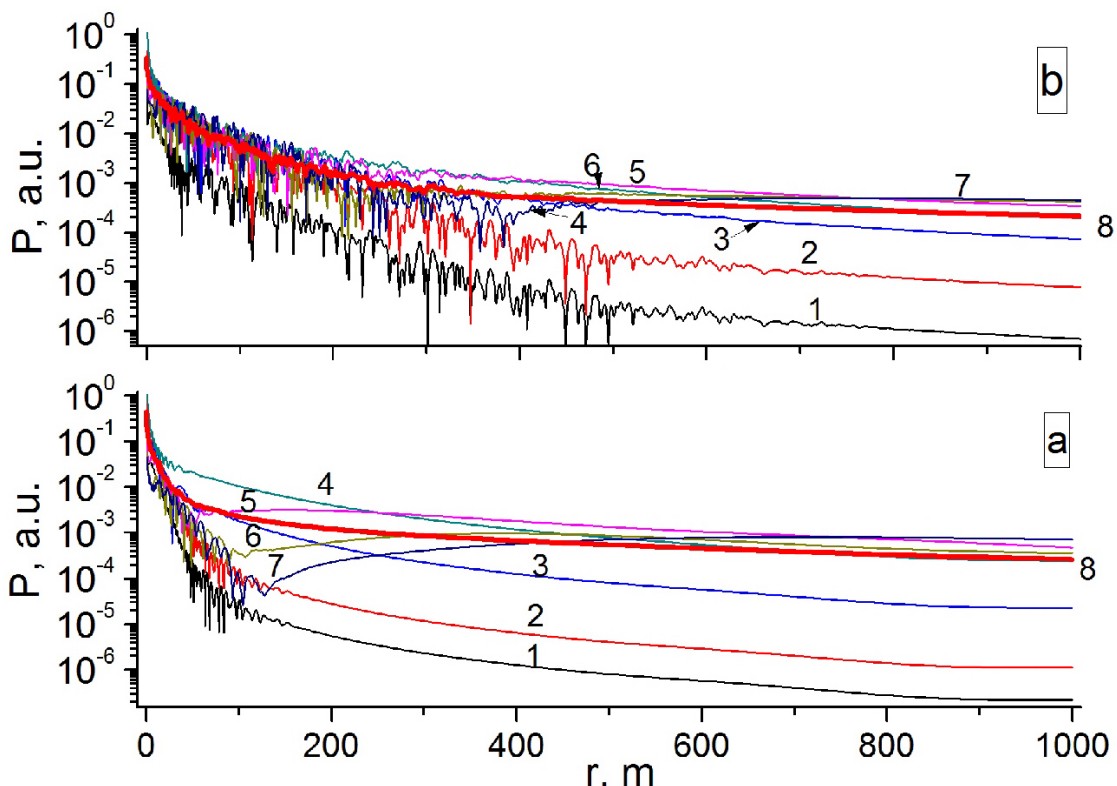

**Figure 14.** Dependences of the acoustic field pressure $P = |p(r, z)|$ in arbitrary units (a.u.) at different bubble concentrations: (**a**) $10^{-6}$, (**b**) $10^{-7}$ and constant layer thickness $h = 7$ m for different depths: 1—$z = 0.1$ m; 2—$z = 0.4$ m; 3—$z = 4.2$ m; 4—$z = 10.5$ m; 5—$z = 21$ m; 6—$z = 31.5$ m; 7—$z = 42$ m. Thick lines 8 show the average depth pressure of the acoustic field $P(r) = <|p(r, z)|>_z$.

Figures 15 and 16 show the dependences on the acoustic field pressure distance at different frequencies $|p(r; f)|$ without a bubble layer and in the presence of a bubble layer. It can be seen that the presence of even a relatively small concentration of bubbles $x = 10^{-7}$ in the layer leads to a significant absorption of sound along the propagation path. At the same time, a significantly stronger frequency dependence is observed compared to the absence of a bubble layer.

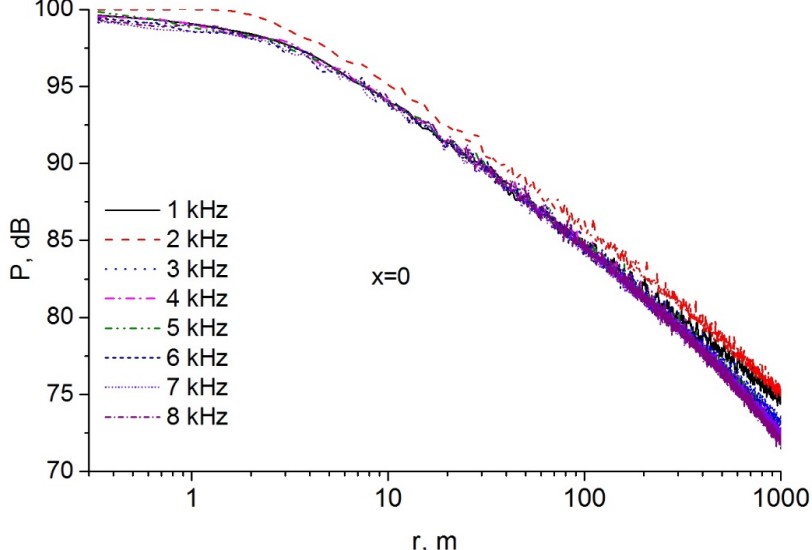

**Figure 15.** Dependences on the acoustic field pressure distance at different frequencies $|p(r; f)|$ without bubble layer.

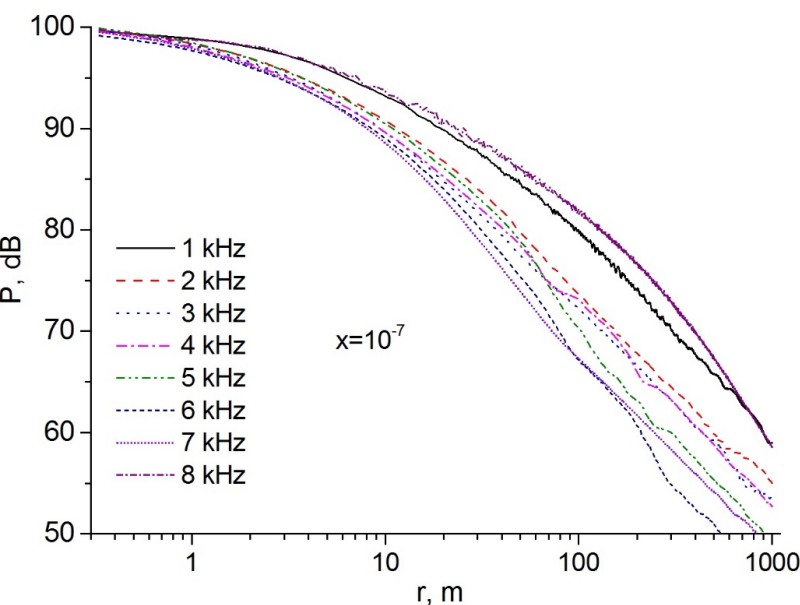

**Figure 16.** Dependences $|p(r; f)|$ at bubble concentrations $x = 10^{-7}$ and constant layer thickness $h = 7$ m.

## 4. Discussion

The above results were obtained in a shallow sea, which is characterized by strong variability of the near-surface layer, as well as the simultaneous content of various micro-heterogeneities. Figures 2–4 show typical results of sound scattering, which show the presence of various objects—bubble clouds and plankton. Their contributions to the scattered signals often intersected, and with the standard procedure for processing acoustic data, it is extremely difficult and often impossible to separate these contributions.

To separate sound scattering on bubbles from scattering on plankton, we used the method of unsteady scattering of pulses of different durations with different frequencies [17,18], which was previously used to study the spectral distribution of bubbles in the near-surface layers of the deep sea. Figure 5 shows a typical application of pulses with different frequencies to obtain information about the structure of bubble clouds and their dynamics for several days. Figure 6, representing the recording of backscattering signals for four minutes, illustrates in detail the variability of the near-surface layer of bubbles formed during the collapse of surface waves and reaching a depth of about 7 m. It should be noted that the data in Figure 6 refer to the autumn-winter period, when the air temperature becomes significantly lower than the water temperature. It is possible that it is under these conditions that facilitated conditions are created for transporting cooled layers of water with bubbles into the sea, despite the rather small wind speed shown in the inset Figure 8. It should be noted that in the experiments shown in Figures 4 and 6, a quasi-homogeneous near-surface layer of bubbles can be observed. The alternation of individual acts of collapse of surface waves in shallow sea conditions roughly coincides with similar ones in open deep sea conditions. It can be assumed that the reason for the overlap and formation of a quasi-homogeneous structure of bubble clouds in shallow sea conditions is the gradual pumping of gas during the collapse of waves. Such aeration of the near-surface layers creates an increased gas content in the water, which prevents the rapid dissolution of bubbles and, under certain conditions, maintains an equilibrium concentration of bubbles in the near-surface layers of the sea.

Experimental data on sound scattering made it possible to present the size distribution of bubbles in an analytical form using a semi-empirical Equation (19). This Equation is in many respects close to the known dependencies proposed in the literature [7,11,13,14]. The advantage of Equation (19) is the practicality and speed of calculating the acoustic characteristics of water with bubbles. Attention should be paid to the presence of a fitting parameter in Equation (19)—the size $R_p$ corresponding to the maximum of the function

$g(R)$. This parameter is important for more accurate calculations. However, its magnitude has not yet been very clearly studied. So from Figure 8 it can be seen that in conditions of moderate wind and a weak layer of bubbles, the value $R_p$ can be estimated at about 10 microns. However, even in conditions of strong wind, the distribution of bubbles in Figure 8 indicates a wider distribution in size and, in particular, indicates a shift of the maximum $R_p$ towards values less than 10 microns. This issue requires further study.

The creation of a quasi-homogeneous layer can lead to important changes in the physical characteristics of the structure of the surface layer of the sea. Particular attention should be paid to the variability of acoustic characteristics—the dispersion of sound velocity and sound absorption. In the case of a quasi-homogeneous layer, their variability will be determined mainly not by individual acts of collapse of surface waves, but by the very average structure of such a layer, which, presumably with a strong wind, can be located under the sea surface at considerable distances. Examples of data processing for studying the variability of the sound absorption coefficient and the volume content of gas in bubbles, in Figures 11 and 12, show the presence of a quasi-homogeneous background near-surface layer. The periods of its variability are much longer than the typical periods of collapse of surface waves.

The modeling of sound propagation in the conditions of such a quasi-homogeneous layer, undertaken in our work, shows that its influence is very significant. It leads both to a change in the laws of the average decay of the sound field along the sound propagation path, and to a change in the shallow spatial structure of the field. It seems important to us to pay attention to the fact that at long distances, despite the presence of a bubble layer, an additional change in the amplitude of the field stops due to the exponential decline caused by the presence of bubbles. The mechanism of such an impact seems to us as follows. The influence of the near-surface layer of bubbles consists in an additional decrease in the field at moderate distances caused by the attenuation of part of the sound energy propagating in the bubble layer. In the future, this energy is completely absorbed, which eventually leads to the absence of the contribution of the bubble layer in the exponential law—only exponential attenuation due to dissipative processes in seawater remains. Nevertheless, the presence of dissipation in the near-surface layer of bubbles can lead to a significant restructuring of the structure of the acoustic field, as demonstrated in Figures 13–16.

## 5. Conclusions

The study examined the structure of the upper layer of the sea saturated with gas bubbles, as well as their relationship with the acoustic characteristics of bubble clouds formed by the collapse of surface waves in a strong wind. Experimental studies were carried out in the shallow sea using the method of non-stationary acoustic spectroscopy, the discussion of which is presented in the paper. The paper demonstrates typical areas in which it is necessary to take into account the joint scattering of sound on bubbles and other inhomogeneities of seawater, where plankton has the greatest contribution. Under these conditions, separation of contributions to sound scattering from bubbles and plankton is required. Otherwise, inflated concentrations of bubbles can be obtained, as well as an incorrect distribution of the concentration of bubbles in depth.

The results obtained by us are new, differing from the works of other authors [2–16], for whom such an analysis has not been carried out. As a result, data on the size distribution of bubbles at various depths have been obtained, which can be described by a power function with exponential decline at small bubble sizes of the order of 10 microns. For practical purposes, for example, for the assessing of the gas exchange between the ocean and the atmosphere, it was important to assess the gas content in bubble clouds under various sea conditions, including storm conditions. Such results were obtained with a sufficiently strong wind, but it is of interest to conduct research at catastrophic speeds when a storm turns into a hurricane. Currently, such data are not available. We hope to use the bottom station in the future to study the state of near-surface layers with bubble clouds during hurricane winds.

In this paper, in order to study the effect of bubbles on the acoustic characteristics of the upper layer of the sea, theoretical estimates of the dependences of the absorption coefficient and the dispersion of the sound velocity on the concentration of bubbles were obtained within the framework of a homogeneous model. It was found that there is a significant excess absorption of sound in bubble clouds, which is hundreds of times higher near the sea surface than the absorption of sound in clear water. It is important that using unsteady sound scattering, it is possible to identify the detailed structure of near-surface bubble clouds, their dynamics and determine the variability of the acoustic characteristics of the near-surface layer under various sea conditions, including the processes of collapse of wind surface waves in a developed storm.

It is shown that when the wind increases in a shallow sea, a quasi-homogeneous near-surface layer of bubbles is observed. Modeling of sound propagation in the presence of a quasi-homogeneous bubble layer shows that it leads both to a change in the laws of the average decay of the sound field along the sound propagation path and to a change in the shallow spatial structure of the field.

**Author Contributions:** Conceptualization, V.A.B. and L.K.B.; Methodology, V.A.B.; Software, L.K.B. and A.V.S.; Validation, V.A.B., A.V.S. and L.K.B.; Formal Analysis, V.A.B.; Investigation, V.A.B., A.V.S. and L.K.B.; Resources, V.A.B.; Data Curation, V.A.B. and A.V.S.; Writing—Original Draft Preparation, V.A.B.; Writing—Review & Editing, V.A.B.; Visualization, V.A.B., A.V.S. and L.K.B.; Supervision, V.A.B.; Project Administration, V.A.B.; Funding Acquisition, V.A.B. All authors have read and agreed to the published version of the manuscript.

**Funding:** The work was carried out with the support of the grant of the Russian Science Foundation No. 22-22-00499.

**Institutional Review Board Statement:** Not applicable.

**Informed Consent Statement:** Not applicable.

**Data Availability Statement:** Not applicable.

**Acknowledgments:** The authors are grateful to I.V. Korskov for technical support during the experiments and S.N. Sosedko for assistance in processing experimental data.

**Conflicts of Interest:** The authors declare no conflict of interest.

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
