# Peer review of "On Sound Scattering and Acoustic Properties of the Upper Layer of the Sea with Bubble Clouds"

_jmse, doi:10.3390/jmse10070872_

Round 1

Reviewer 1 Report

The manuscript "On sound scattering and acoustic properties of the upper layer of the sea with bubble clouds" investigates the effects of the bubble clouds on the acoustic characteristics of the upper layer of the sea. Experimental studies were carried out in the shallow sea using the method of non-stationary acoustic spectroscopy to obtain the size distribution of bubbles at different depths. The abnormal sound scattering in the shallow sea has been well studied. However, there are some revisions should be made before publication.

  • Line 81, “…in the form of [1] in the single scattering approximation”, this statement should be rephrased to be understood.
  • Some of the formulas contains two equations, such as (1) (4) (7) (8) (9) (12). It is difficult to follow which equation is referred when you mention a specific formula. The second one seems to be the supplement of the first one.
  • Some of the results seems to be published in other papers of the authors, such as Figure 2. In this case, the reference should be appropriately cited.
  • In the experiment setup, a non-directional broadband hydrophone was installed for recording, calibrating and monitoring radiation. However, the information of the hydrophone is missing, including its bandwidth, sensitivity, placement.
  • Line 320, it seems there are some typos for “L~(2÷4)” “m=1÷3” “n =3.3÷3.8”.
  • The result and discussion seem to be highly related. These two parts should be merged and reorganized.
  • The overall quality of the figures is poor. High resolution images should be replaced.

Author Response

Thank you for reviewing the manuscript.

The answer is in the file: The answer for Review 1_ jmse-1730364.doc

Reviewer 2 Report

Section 1 must be improved. Authors should emphasize contribution and novelty, the introduction needs to clarify the motivation, challenges, contribution, objectives, and significance/implication.  You must properly introduce your work, specify well what were the goals you set yourself and how you approached the problem. At the end of the section, add an outline of the rest of the paper, in this way the reader will be introduced to the content of the following sections.

Section 2 must be improved. In the first part of the section, you present the theoretical discussion. In this section you make extensive use of equations that are often not properly introduced. You must properly introduce the equation, list in detail the variables contained in it with a concise description of the meaning. To make them more readable show them in a bulleted list. In this way the reader will be able to understand the contribution of each variable. In subsection 2.2 then present graphs with results. it is still early to present data, move it to the next section. In subsection 2.3 then you present the experimental part of your work. Unfortunately you devote little space to it, but I believe it should be treated in greater detail. Describe in detail the equipment used to make the measurements. Extract this data from the datasheet of the instrumentation manufacturer. To make reading the specifications of the instruments more immediate, you can insert them in a table, listing the instruments used and the specific characteristics for each. Make extensive use of equipment, some of which lie on the seabed. Where are the images of the experimental set up of your work. This gives a considerable thickness to your experimental work.

Section 3 must be improved. In this section you present the results. Furthermore, a description of the hardware and software used for data processing is completely missing. Describe in detail the hardware used:  Extract this data from the datasheet of the hardware manufacturer. To make reading the specifications of the hardware more immediate, you can insert them in a table, listing the instruments used and the specific characteristics for each. Also you should describe in detail the software platform you used. Also describe the libraries you used. Make extensive use of color-map figures: You should first improve the quality of the images, they often appear blurry. Also you should clarify in the captions the variables that are represented with colors and define the unit of measurement. Then some figures are confused (for example in figure 6) there are many diagrams without labels, in this way the reader is confused.

Section 5 must be improved. Paragraphs are missing where the possible practical applications of the results of this study are reported. What these results can serve the people, it is necessary to insert possible uses of this study that justify their publication. They also lack the possible future goals of this work. Do the authors plan to continue their research on this topic?

81)” in the form of [1]” If you refer to equation (1) use the ()

82) “Born approximation” Introduce adequately the topi and add references to allow the reader to learn more about the topic

83) Where is mv and Pi explanation?

84-87) List the variable explanation as bullet list

89) Where is Pbs explanation?

240) Figure 3 offers a schematic description of your work. It would be advisable that you also add real images of the equipment used. Add image of the bottom station

280) Figure 6, add label to subplot and explain it in the caption

285) Figure 7, add label to subplot and explain it in the caption

301) Figure 8, add label to subplot and explain it in the caption

310) Figure 10, add label to subplot and explain it in the caption

342) Figure 11, add label to subplot and explain it in the caption

366) Figure 12, add label to subplot and explain it in the caption

389) Figure 13, add label to subplot and explain it in the caption

401) Figure 14, add label to subplot and explain it in the caption

Author Response

Thank you for reviewing the manuscript.

The answer is in the file: The answer for Review 2_ jmse-1730364.doc

Round 2

Reviewer 2 Report

The authors addressed only partially the reviewer's comments with attention and modified the paper with the suggestions provided. Some suggestions proposed by the reviewer were contested by the authors, even if in some points they modified the paper in the sense suggested. I refer to the equations that at least in the initial part have been revised and are now easier to read. The same strategy should also be adopted for the remaining equations. The purpose of a paper is to disseminate one's work, the more readable the work is and the more people will read it. In this way, both the author and the journal will benefit. Anyway, the new version of the paper has improved both in the presentation that is now much more accessible even by a reader not expert in the sector, and in the contents that now appear much more incisive.

Minor revision:

- Try to enrich the captions of the figures, the reader should be able to read the figure without the need to retrieve the information in the paper. Try to summarize the essential parts of the Figure and what you want to explain with it.

- Make sure that the legends are placed outside the graphs, otherwise they cover part of the diagram. For example in the case of Figure 7.

- When in a Figure you show multiple diagrams label with a letter (a, b, c, ...) and then add a description for each subplot in the caption. For example, Fig. 4,5,7,8, 11, 13,14

321) In Fig. 2 legend a value is wrong (1633 or 16330 ?)

424) Explains in the caption the reason for his graph that highlights a portion of the graph.

435) Add legend to Figure 10

Author Response

We are grateful to the reviewer for a careful reading of the article and very useful comments. Please see the attachment.
